# A New Magnesium Phosphate Cement Based on Renewable Oyster Shell Powder: Flexural Properties at Different Curing Times

**DOI:** 10.3390/ma14185433

**Published:** 2021-09-20

**Authors:** Hui Wu, Zhujian Xie, Liwen Zhang, Zhiwei Lin, Shimin Wang, Wenle Tang

**Affiliations:** Department of Civil Engineering, Guangzhou University, Guangzhou 510006, China; Whi1223@163.com (H.W.); zhujianxie@163.com (Z.X.); linzhiwei9801@163.com (Z.L.); wangshimin445311@163.com (S.W.); 13437890351@163.com (W.T.)

**Keywords:** magnesium phosphate cement, flexural strength, oyster shell powder, curing time, three-point bending test

## Abstract

Magnesium phosphate cement (MPC), a new type of inorganic cementitious material, is favored in engineering and construction because of its fast setting speed and high bonding strength, but is limited in practical application due to its high production cost and excessive release of hydration heat. Relevant research has investigated the application of discarded oyster shell powder (OSP) replacing cement mortar and has reported certain improvements to its performance. Consequently, focusing on discovering more effects of OSP on MPC performance, this study, by using a typical three-point bending test, used 45 cuboid specimens to investigate the influences of OSP mass content on flexural properties of MPC at different curing times. Results illustrated that MPC flexural strength was first increased and then decreased, and 3% is the critical value for OSP mass content. Similarly, the stiffness of all specimens presented a tendency to increase first and then decrease, with a maximum value of 36.18 kN/mm appearing at 3%, i.e., the critical OSP mass content. Finally, scanning electron microscope (SEM) and X-ray diffraction (XRD) were employed to analyze the microstructure and composition of specimens, confirming that the specimens generated not only the hydration product potassium phosphate magnesium (MgKPO_4_·6H_2_O, MKP), but also another new reactant (CaHPO_4_·2H_2_O).

## 1. Introduction

As a new type of inorganic cementitious material, magnesium phosphate cement (MPC) is manufactured by first mixing magnesium oxide, potassium dihydrogen phosphate, fly ash, and borax proportionally and then reacting the mixture with water [1,2]. Because of its excellent performance in early strengthening and rapid hardening, as well as its strong bonding force, marvelous volume stability, high heat and temperature resistance, broad compatibility with other concrete, long durability and friendly environmental adaptability [3,4,5,6,7], it is widely used in repairing medium fields such as roads, bridges and dams [6,8,9]. However, as for magnesium oxide, the fundamental raw material of MPC, its manufacturing requires a highly standardized process and a great amount of production cost, and usually releases an excessive hydration heat [10], consequently hindering MPC’s practical application in engineering construction. Therefore, a substituting material for MgO to fill into MPC is quite necessary and needed.

Oyster shell powder (OSP) is a green, environmentally friendly, and renewable powdered solid, of high absorption and durability, which is produced by grinding the raw material of oyster shell after sterilization at room temperature, and which contains about 95% calcium carbonate [11]. Previous research has proved that applying OSP as a supplementary cementitious material in concrete helps improve the workability, compressive strength, freeze–thaw resistance, water permeability, and durability of concrete [12,13,14,15,16,17]. In view of this, this study used disposed oyster shell powder as an admixture to replace part of MgO in preparing MPC, expecting to overcome the above-mentioned drawbacks of MPC, and thereby contributing to energy saving, consumption and cost reduction, and resource recycling and comprehensive utilization. For designing and applying OSP–MPC scientifically and reasonably in actual engineering construction, research on OSP–MPC’s mechanical properties is vitally necessary. Thus, this research focused on the influence of different OSP mass contents on MPC flexural properties by including 45 specimens in three groups corresponding to the curing times of 7, 14, and 28 days. Three-point bending test (TPBT) was adopted to find out the influence law of various OSP mass contents (0%, 3%, 6%, 9%, 12%) on the failure mode, flexural strength, load-displacement curve, and compression ratio of MPC. In addition, modern micro-testing techniques such as scanning electron microscope (SEM) and energy dispersive X-ray detector (EDX) were used to identify the changes in the microstructure and hydration composition of MPC at different curing times after the incorporation of OSP, and to determine the microstructure and composition of the product, to explain the properties microscopically.

## 2. Experimental Programs

### 2.1. Test Specimens

A total of 45 specimens with a size of 40 mm × 40 mm × 160 mm in this TPBT test were divided into 15 sets in reference to GB/T 17671-1999 [18], as shown in Figure 1. Following the curing times of 7, 14, and 28 days, 15 sets of specimens were subdivided into 3 groups, with each group including 5 sets according to their OSP mass content. Therefore, three specimens in the same set were of identical parameters and named by the order of test types (influencing factors and their values). For example, M-SW3-T7-1 means the first specimen in the set with an OSP mass content of 3% which has undergone a bending test at a curing time of 7 days. More detailed information of all specimens is presented in Table 1, where D represents the dosage of OSP (mass content). In addition, specimens in sets M-SW0-T7, M-SW0-T14, and M-SW0-T28 were designated as reference specimens since they meant the specimens without OSP.

The preparation of the OSP–MPC specimen is shown in Figure 2. First, a mixture consisting of potassium dihydrogen phosphate (KH_2_PO_4_), borax, and fly ash (FA) in a certain proportion was poured into a cement slurry mixer and then stirred after tap water was added to produce a uniform slurry. Then, magnesium oxide (MgO) and a certain amount of OSP substituting for MgO were added into the mixer and stirred at low speed for 30 s and then at high speed for 90 s until the MPC slurry thoroughly interacted. Next, the mixed MPC slurry was poured into the mold for a complete solidification (after about 30–60 min); specimens were thus developed and demolded. Finally, all specimens were ready after being naturally cured to their corresponding time. It should be noticed that all specimens must be polished and sprayed with white paint on the outer surface to facilitate the observation of failure process before the bending test.

In reference to the standard ASTM-C1609 [19], specimens were first tested by a universal testing machine shown in the picture, and then polished and sprayed with white paint (see in picture) to mark their failure process. Each specimen was placed between two hinged supports, its edge 30 mm from each support, and its center distance kept in 100 mm, as shown in Figure 1.

As the test initiated, first, a preload at a speed of 2.0 mm/min based on the displacement of the universal testing machine was added to the machine until it reached 0.02 kN, and then data began to be recorded. While the specimen remained under load at a speed of 1.0 mm/min, its load-displacement curve was recorded by a load-bearing sensor and its failure process was recorded by a high-definition camera during the bending test. The test was terminated when specimen’s residual load reached 10% of the peak load. In addition, some failed specimens were used for SEM, EDX, and XRD tests to observe specimens’ microstructures and hydration products under different dosages of OSP, so as to facilitate the interpretation of the observations in the bending test.

XRD was applied to analyze sample elements and compound content changes. The basic process for results of X-ray diffraction effects on materials uses sample atoms to show the primary microscopic ion excitation, i.e., X-ray photons produce secondary X-ray and X-ray diffraction patterns. We used Jade software after analyzing the diffraction patterns and sample composition spectrum diffraction peaks. Preparation of XRD samples: the samples are ground into powder by ball mill, the powder is ground through 200 target screening, to achieve the fineness of the test sample, sealed and preserved; XRD test process: fill the central groove of the glass slide with OSP powder; use cover glass to smooth the protrusion of powder; use a small brush to clean the residual powder at the edge of the groove; put into X-ray diffraction chamber for 5°~80° wide angle test.

### 2.2. Material Properties

The mixing proportion of all specimens shown in Table 2 align with previous relevant studies [20,21,22]. Re-burning (1500 °C) was provided by Zhengyang Foundry Material Factory, Xinmi City, Henan Province, China, with a specific surface area of 227.5 m^2^/kg, density of 2650 kg/m^3^, and particle size of 45 μm. Detailed chemical composition is listed in Table 3. Potassium dihydrogen phosphate and borax were acquired from Jianghua Chemical Glass Co., Ltd., Nanjing City, Jiangsu Province, and both have the same particle size of 350 μm. Potassium dihydrogen phosphate is stable in air, hygroscopic, soluble in water but insoluble in alcohol, and borax is a white crystal powdery substance at room temperature, odorless and salty. Before mixing with MgO, potassium dihydrogen phosphate and borax were first dried in an oven for at least 24 h, and then ground in a ball mill for 2 h, respectively. As presented in Table 3, FA of American Society of Testing Materials (ASTM, West Conshohocken, PA, USA) C618 class F was employed to replace part of MgO to improve the workability of MPC mortar, with a particle size of 43 μm.

The oyster shell powder used in this project was obtained by grinding discarded oyster shells into powder. Its XRD analysis is shown in Figure 3 and its physical properties are listed in Table 4, and it was provided by Lingshou County Ruixing mineral powder factory, ShiJiaZhuang City, HeBei Province, China. Before grinding, oyster shells were washed until the attached sludge and water weeds on the surface of oyster shells were rinsed, and then they were initially fragmented by a jaw crusher after natural drying. Then, the crushed shell fragments were taken for their first grinding through a light ball tube mill, and then the second grinding through an experimental ceramic ball mill. Finally, OSP passed through the vibration screen until they were filtered to a fineness of 325 mesh with a sieve residue of less than 6%. Figure 4 explains the overall process of OSP.

## 3. Failure Process and Modes

Figure 5a–c respectively show specimens’ failure modes under three curing times, and display an identical failure mode for specimens in each group when the load was added to the peak load of flexural strength.

For specimens in Group T7 with no OSP (Group OSP0-T7), typical brittle failure emerged. Shortly after loading, tiny splits appeared in the center bottom of the specimen, with no obvious failure. Then, the splits rapidly stretched across the specimen, bringing about a crisp cracking sound within seconds and a brittle failure with two parts still attached to each other. Brittle failure showed up when OSP was incorporated and grew more apparent as OSP mass content increased. For example, as shown in Figure 5a, for OSP6-T7-OSP12-T7, the increase of OSP mass content resulted in the gradual expanding in the width of the crack’s lower edge and a crisp cracking sound. When the crack reached out even wider, the specimen was broken into two segments. Furthermore, the deflection of the specimen was greater than that in CF0-T7, which ascended with the OSP mass content, as shown in the load-deflection (L-D) curve in Figure 9a.

With the same OSP mass content, specimens in Group T14 and T28 presented a similar failure process to that of Group T7. Specimens in Group T14 did not break into two parts when they were loaded to their limit, and their width of the mid-span crack at the bottom section was less than that of the specimen with no OSP. As for the specimens in Group T28, when the ultimate load was added, the width of the mid-span crack at the bottom section was more overt than that in the Group T7 and T14. 

In addition, the cross-interruption surface of the specimen after its failure was intercepted as shown in Figure 6. For the specimens with OSP mass content less than 12% (OSP0-T7-OSP9-T7), their middle span displayed a relatively flat cross section, whereas the specimens with 12% OSP mass content (OSP12-T7) displayed an irregular and uneven fracture surface. An image analysis technique was used to study the hole area of the fracture section of the specimens with different OSP contents at different curing times. In the process of image analysis, a vacuum cleaner was used to clean the powder to prevent cavity blockage. Then a high resolution digital camera was fixed to photograph the fracture surface of the specimen. Then, image analysis software Image Pro Plus was used to binarize the images taken, and pore structure parameters were analyzed to obtain the area ratio of pore structure, hole area to cross section area (40 mm × 40 mm), as shown in Figure 7. The width and height of the images are 4016 and 6016 pixels, respectively, and the horizontal resolution and vertical resolution are both 300 dpi. The measurement area is the cut-off surface of the specimen (40 mm × 40 mm), and the measured aperture diameter is about 0.5 mm to 5 mm. As for the holes in the specimens, area ratio first decreased then increased with the increase of OSP mass content, among which, the specimens with 3% OSP mass content (OSP3-T7) had produced fewer pores and the best molding pattern. Comparing with Group T7, it was found that when OSP mass content was kept constant, the area ratio in specimens’ damaged sections lessened as curing time prolonged. This is because the generating of internal hydration products in the specimens in Group T7 and T14, which created more pore defects than those in Group 28, in cooperation with OSP partly participating in the internal reaction and filling in the pores, enabled specimens to be completely cured at the curing time of 28 days. Therefore, specimens in Group T28 exhibited more condensed and uniform cross sections. As for the area ratio in fracture sections with different dosages of OSP at the same curing time, taking T28 as an example, it can be found that the area ratio of specimens with appropriate addition of OSP (3%) decreases by 44.9% compared with that without addition of OSP. Then, with the increase of OSP dosages (6%, 9%, 12%), the area ratio continues to increase. The area ratios are 4.50%, 5.69% and 9.67%, respectively. This is because of the decrease in water–binder ratio of the specimen.

## 4. L-D Behaviors

### 4.1. L-D Curves

Figure 8 shows the L-D curves of the specimens. As shown in Figure 8a, L-D curves of specimens all consist of an ascending phase (O–A) and a descending phase (A–B), in correspondence with the L-D curves observed in the bending test. In the ascending phase, load climbed almost linearly to the peak value (point A) as displacement arose, while in the descending phase, all specimens almost demonstrated a linear decline close to zero (point B), showing significantly apparent brittle failure.

As mentioned earlier, specimens in Group T7 increased their loading load to the peak point (point A) in an approximately linear way, and then dropped it sharply to point B. The peak displacement and secant stiffness of specimens are shown in Figure 9a,b, respectively. As OSP mass content ascended from 0% to 12%, peak displacement advanced from 0.101 mm to 0.161 mm, with the maximum increment being 59.4% (0.06 mm) of the original value. Compared with peak displacement, specimen’s Ks tended to rise first and then fall. When OSP mass content expanded to 3%, Ks increased to its maximum value of 22.48 kN/mm, 6.09% more compared to that with no OSP, and then sequentially declined at OSP mass content greater than 3%. Specimens exhibited a similar peak load of increasing first from 2.14 kN (M-SW0-T7) to 2.81 kN (M-SW3-T7) and then dropping to 1.53 kN (M-SW0-T7). This is because the existing tiny cracks inside MPC expanded and caused failures. At OSP mass content of 3%, OSP could better diffuse in internal MPC to fill up pores and provide nucleus for hydration, as well as condensing the internal matrix. However, at OSP mass content greater than 3%, although more pores were filled up by OSP, peak load still kept declining continuously because the reduction of cementitious material incurred a relatively small output of MKP.

Specimens in Group T14 and T28 displayed a similar trend in L-D curves to that of Group T7, but differed in Ks, i.e., for a longer curing time, the Ks of specimens at OSP mass content ranging from 6% to 12% decreased less than that of 0% of OSP mass content. Taking Group T28 as an example, when OSP mass content increased from 6% (M-SW6-T28) to 12% (M-SW12-T28), secant stiffness decreased by 25.7%, 46.2%, and 53.2%, respectively compared to 0% of OSP mass content (M-SW0-T28). However, for Group T7, compared to OSP mass content of 0% (M-SW0-T7), secant stiffness reduced by 30.4%, 46.7%, and 55.2%, respectively as OSP mass content increased from 6% (M-SW6-T7) to 12% (M-SW12-T7). In addition, curing time similarly affected the relationship between peak displacement and OSP mass content. On the contrary, the gap in secant stiffness between specimens in Group T7 and Group T14 decreased with the prolonging of curing time, as shown in Figure 9. For specimens with OSP mass content of 3% and 6%, the gap of peak displacement also descended with the ascending of curing time, as shown in Figure 9. For specimens with the same OSP mass content, the secant stiffness of specimens increased with curing time due to further hydration. In addition, when OSP mass content was kept constant, the slope in the rising part (O–A) indicated an increasing trend for specimens in each group as curing time increased.

### 4.2. Flexural Strength

The flexural strength of a specimen is calculated using formula (1) according to ASTM C293 [23], and finally the average flexural strength of three specimens was taken as the measurement for flexural strength (with accuracy to 0.01 MPa). If any value in these three strengths exceeds the average value by ±10%, the average value is calculated after excluding the exceeding flexural strength.
(1)$$R_{f}=\frac{1.5P_{u}L}{bh^{2}}$$
where *R_f_* is the flexural strength, *P_u_* is the peak load, *L* is the supporting distance (100 mm), *b* is the width of specimen (40 mm), and *h* is the height of specimen (40 mm).

Figure 10a–c describe the peak load and flexural strength of Group T7, T14, and T28, respectively. As OSP mass content increased, the flexural strength of T7 group ascended first and then descended. When OSP mass content was 3% (M-SW3-T7), the flexural strength climbed up to the maximum value of 6.59 MPa, 31.3% higher than that of the reference group (M-SW0-T7). Subsequently, the flexural strength decreased with the increase of OSP mass content. As OSP mass content increased from 6 to 12%, the flexural strength dropped to a value lower than that in M-SW0-T7. Particularly for M-SW9-T7 and M-SW12-T7, their flexural strengths decreased by 22.9% and 28.7%, respectively, compared with M-SW0-T7.

For specimens in Group T14 and T28, OSP mass content had a similar effect on their flexural strength. Flexural strength rose as OSP mass content increased to 3%, and then dropped as OSP mass content continued its augmenting. Therefore, it can be concluded that 3% of OSP is the most appropriate dosage to improve MPC bending performance, while excessive OSP would impair it. This might be due to the decreasing amount of cementitious materials. When OSP mass content was unchanged, its flexural strength increased significantly with the extension of curing time. In addition, no regularity was found in the influence of curing time on the compressive strength with the increase of OSP mass content.

## 5. Microstructure and Hydration

SEM was used to study the influence of OSP mass content on the microstructure of MPC. The SEM sample preparation process causes errors which affect the SEM images [24]; after screening, the results of SEM for specimens at three curing times are displayed in Figure 11a–c, respectively, in which Group T7, T14, and T28 showed similar microscopic trends. For example, in Group T28, when OSP mass content was 0%, some pore defects and microcracks were found in the matrix. As OSP mass content increased to 3%, a small amount of existing filler in the voids supported the pore sidewalls and halted the spreading of micro-cracks. As OSP mass content continued increasing, the MPC matrix gradually decreased its density and increased the progression of microcracks, resulting in more internal defects. This is because the decrease in water–binder ratio of the specimen had a much greater impact than that of the specimen with a 3% of OSP mass content, leading to a decline in its flexural strength. It proves that a proper amount of OSP filling in MPC could contribute to improving the bending resistance of MPC. Comparing Group T7 with Group T14, it is clear that when OSP mass content is kept constant and curing time increases, every specimen would develop more thoroughly, the rod-like crystal structure would gradually decrease, and the network structure would increasingly condense. As a result, the flexural strength of specimens in each group increased with curing time. The overall variation trend of pore defects and microcracks with the mass content of OSP shown by SEM images is consistent with the development trend of pores in fracture sections shown in Figure 6. The development trend of pores in groups T7, T14, and T28 is similar. With the increase of OSP content, the size and distribution of cross sections of pores first decrease and then increase. When the OSP content is 3%, the distribution of cross sections of pores is the least. With the growth of age and the development of hydration, the pore defects are gradually reduced.

In order to further analyze the composition of the filling material inside the MPC pores after adding OSP, XRD analysis was carried out. Figure 12a–c describe the XRD results of specimens at 7, 14 and 28 curing days, respectively. Taking Group T28 as an example, when OSP mass content was 0%, there were mainly overreacted MgO and hydration product MKP in MPC, the intensity of MPK first ascended from 894 a.u. to 1473 a.u., and then descended by 745 a.u. as OSP mass content increased from 0% to 12%. In addition to CaCO_3_, the main component of OSP, CaHPO_4_·2H_2_O was detected after the incorporation of OSP, with OSP mass content increasing from 3% to 12% and its maximum intensity dropping from 1318 a.u. to 599 a.u. This phenomenon may be ascribed to two reasons. The first reason is the hydration of MPC [25,26]. KH_2_PO_4_ was dissolved into potassium ion and dihydrogen phosphate ion, and MgO was converted into hydroxyl ions (OH^−^) and magnesium ions (Mg^2+^) through the interaction of free water, therefore MKP was generated by Mg^2+^ finally interacting with K^+^, PO_4_^3^^−^ and bound water from H^+^ and OH^−^, as shown in Figure 13. The second reason is that since OSP was incorporated into MPC as an alternative of part of MgO, calcium ions (Ca^2+^) were dissolved from CaCO_3_ in OSP under the assistance of free water, as shown in the figure, and then reacted through combining with the remaining phosphate ions (PO_4_^3−^) in the matrix to produce CaHPO_4_, as shown in Figure 14. Therefore, the generated CaHPO_4_ could fill in the internal pores and enhance the matrix strength. Matrix density was also well promoted through the interaction of hydration products MKP and CaHPO_4_, as shown in Figure 15. In addition, when OSP mass content was greater than 3%, the increasingly replaced MgO would result in the gradual decrease of the remaining phosphate, which inhibited the dissolution process of MKP and phosphate in OSP–MPC hydration products and reduced the content of MKP and CaHPO_4_·2H_2_O, finally degrading the matrix strength. In the comparison of Group T7 with Group T14, their variation trends were similar to that of T28 group, and it was discovered that the diffraction peak intensity of MKP and CaHPO_4_·2H_2_O gradually increased as the curing time increased. Therefore, specimens in Group T28 exhibited the highest flexural strength.

## 6. Conclusions

In this study, a typical three-point bending test was employed to discover the influence of OSP mass content on the failure mode, flexural strength, and deflection behavior of MPC under different curing times. In addition, analysis techniques of SEM and XRD were adopted to study the new reaction products and morphology of OSP–MPC cement paste. In view of the results, the following conclusions can be drawn:(1)Limestone has similar compositions and properties to OSP, and the strength of MPC can be slightly increased by partially replacing magnesia in MPC with 5% limestone, while the strength of MPC decreases as the limestone content continues to increase [27]. In this paper, a certain amount of OSP can help improve the pore structure of MPC. When 3% of OSP is mixed with MPC, the specimen can effectively halt the spreading of cracks and alleviate its brittle failure.(2)OSP mass content has an identical effect on the bending properties of MPC at different curing times. However, specimens at 28 curing days exhibit greater flexural strength and higher stiffness than those at 7 days and 14 days.(3)OSP can improve the bending strength to a certain extent, which is however confined by its mass content. When OSP mass content is greater than 3% (in this study), flexural strength will decrease. Further research is needed to obtain a more accurate critical value for OSP mass content.(4)SEM analysis proves that as curing time increases, the development of specimens is relatively complete, the rod-like crystal structure gradually weakens, and the network structure gradually becomes dense. As a result, the strength of specimen gradually increases.(5)XRD analysis demonstrates that in addition to the main hydration product struvite and the residual magnesium oxide, a new reactant (CaHPO_4_·2H_2_O) is also generated, and pore structure in the matrix is optimized to interact with MKP, which mutually improve the mechanical properties of MPC.

## Figures and Tables

**Figure 1 materials-14-05433-f001:**
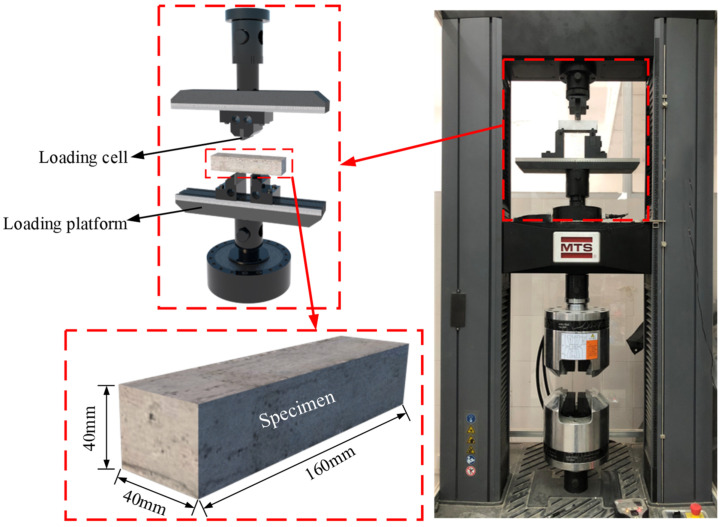
Test setup and specimens.

**Figure 2 materials-14-05433-f002:**
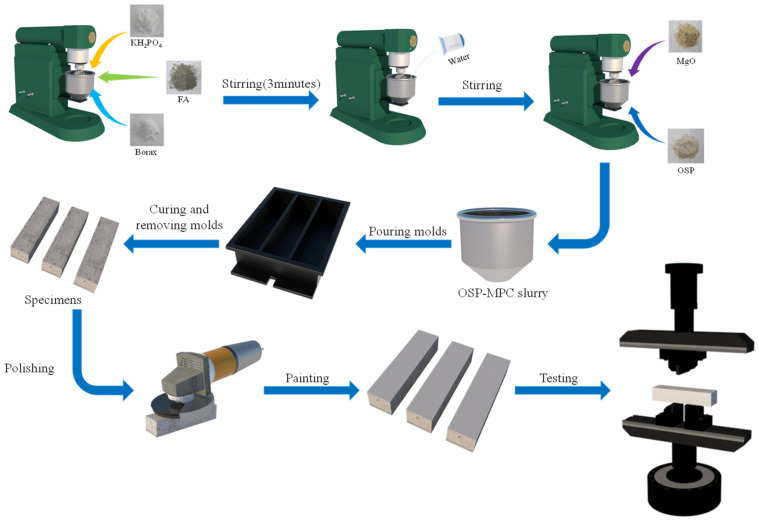
Specimen preparation.

**Figure 3 materials-14-05433-f003:**
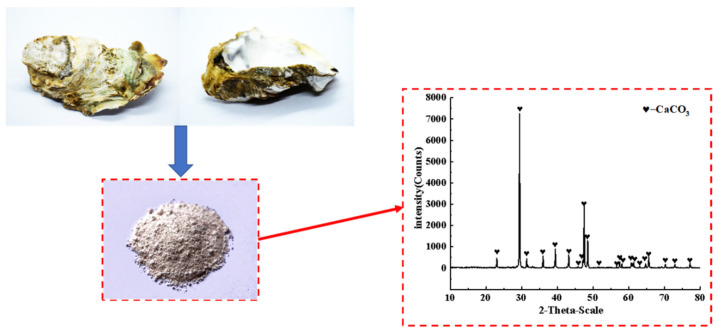
Discarded oyster shells and X-ray diffraction (XRD) analysis.

**Figure 4 materials-14-05433-f004:**
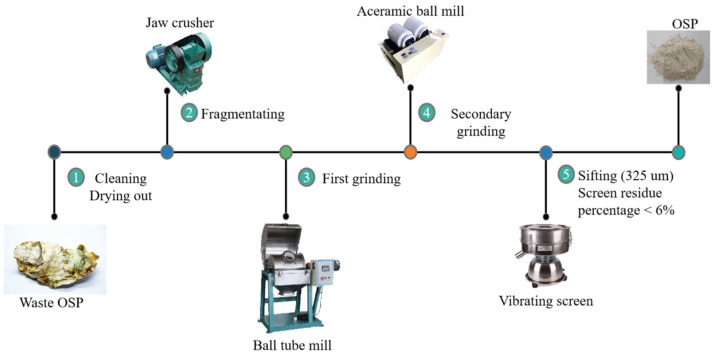
Preparation process of OSP.

**Figure 5 materials-14-05433-f005:**
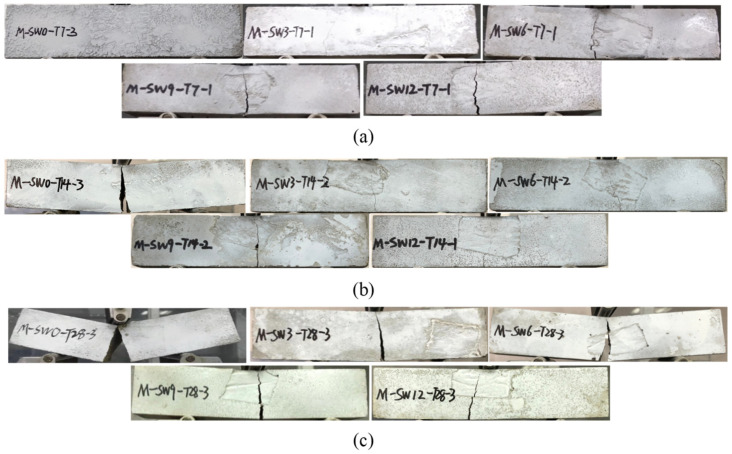
Failure modes. (**a**) T7; (**b**) T14; (**c**) T28.

**Figure 6 materials-14-05433-f006:**
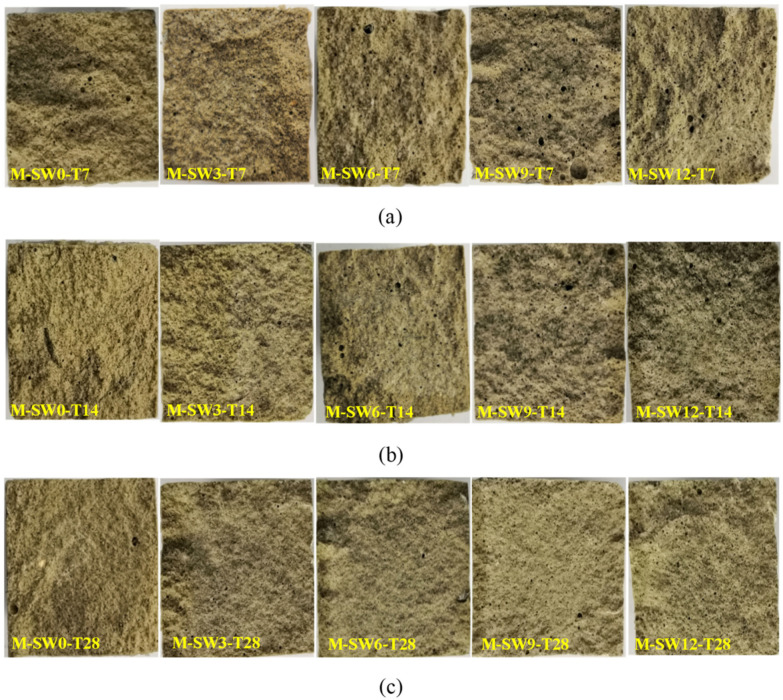
Section of failed specimens. (**a**) T7; (**b**) T14; (**c**) T28.

**Figure 7 materials-14-05433-f007:**
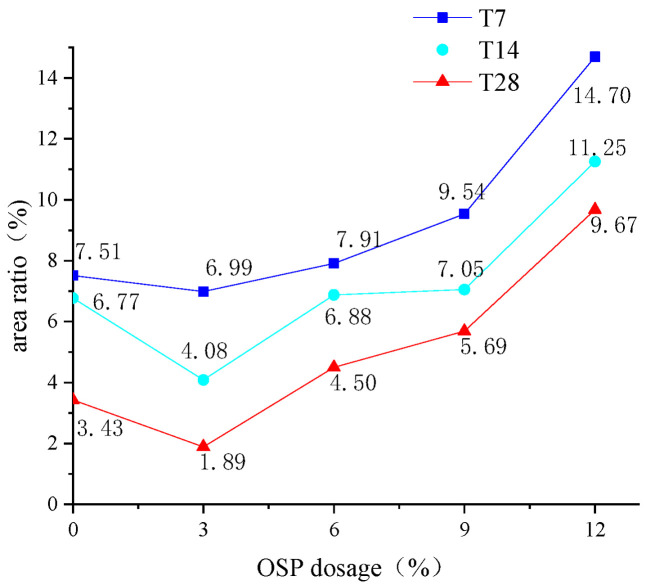
Area ratio of sections.

**Figure 8 materials-14-05433-f008:**
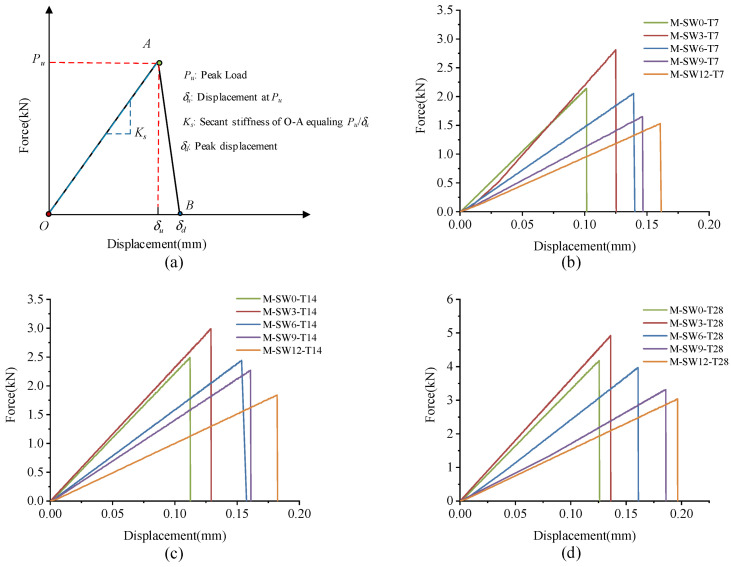
L-D curve of specimens. (**a**) L-D curve configuration; (**b**) T7; (**c**) T14; (**d**) T28.

**Figure 9 materials-14-05433-f009:**
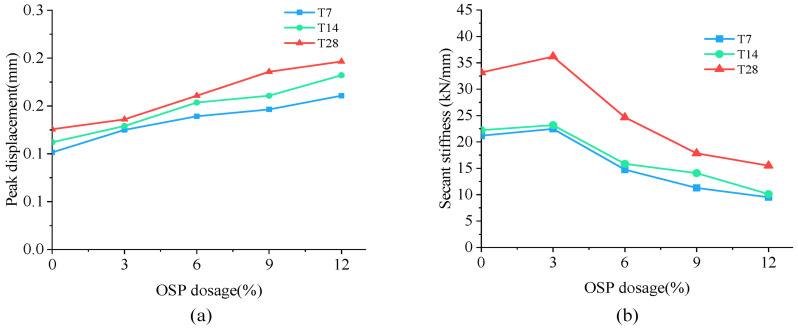
Peak displacement and secant stiffness. (**a**) Peak displacement; (**b**) secant stiffness.

**Figure 10 materials-14-05433-f010:**
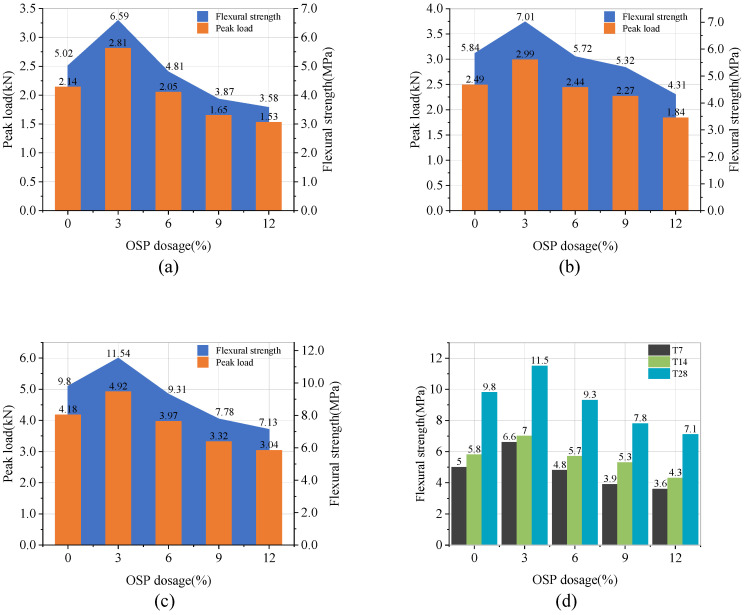
Flexural strength and peak load. (**a**) T7; (**b**) T14; (**c**) T28; (**d**) summarized value of three groups.

**Figure 11 materials-14-05433-f011:**
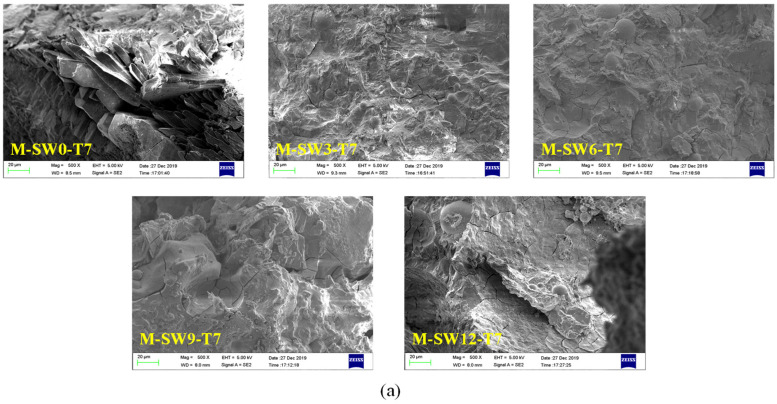
Scanning electron microscope (SEM) analysis of specimens. (**a**) T7; (**b**) T14; (**c**) T28.

**Figure 12 materials-14-05433-f012:**
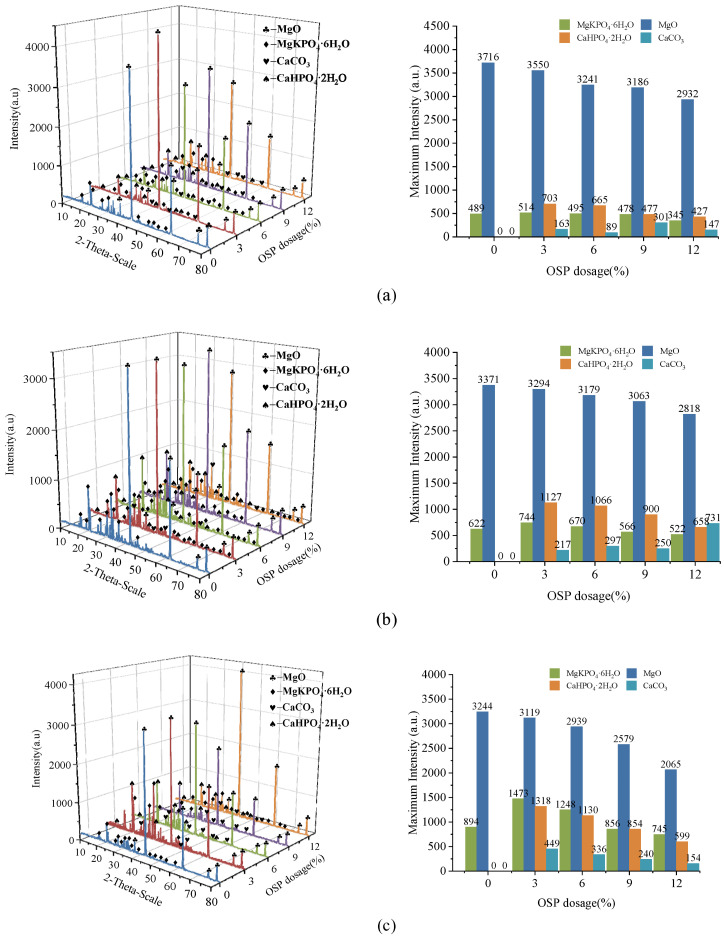
X-ray diffraction (XRD) analysis of specimens. (**a**) T7; (**b**) T14; (**c**) T28.

**Figure 13 materials-14-05433-f013:**
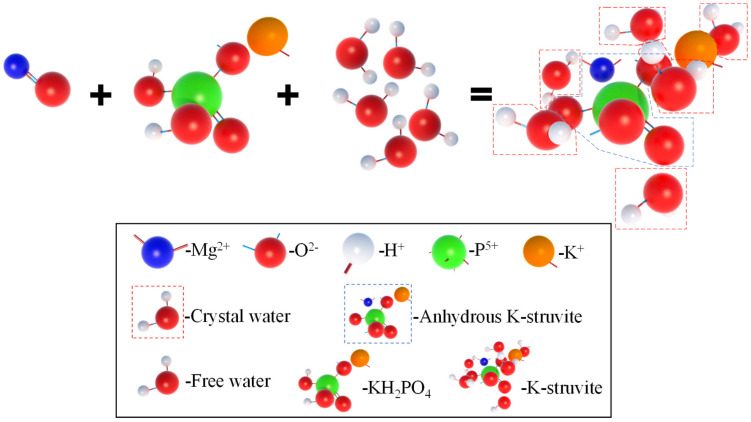
Hydration reaction process of MPC.

**Figure 14 materials-14-05433-f014:**
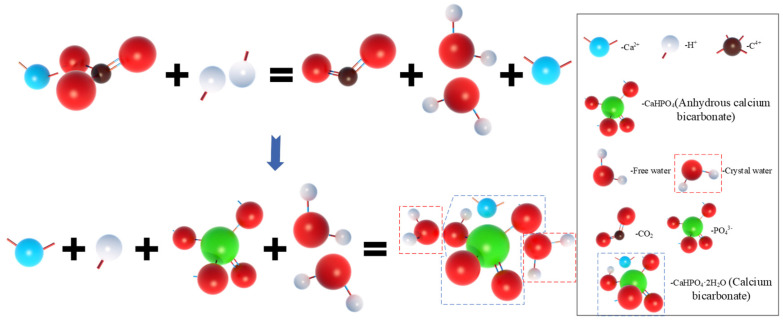
Reaction process of CaHPO_4_·2H_2_O.

**Figure 15 materials-14-05433-f015:**
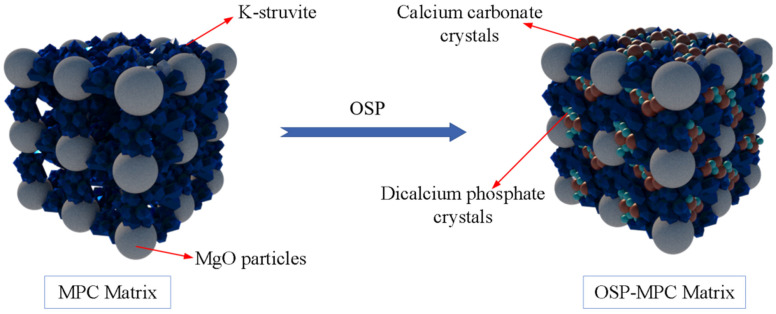
Reaction model of oyster shell powder–magnesium phosphate cement (OSP–MPC) matrix.

**Table 1 materials-14-05433-t001:** Specimen details.

Group	Set	Specimen Number	Curing Time (Day)	OSP		MgO
				D ^1^ (%)	Mass (g)	Mass (g)
T7	M-SW0-T7	M-SW0-T7-1	7	0	0	300
		M-SW0-T7-2				
		M-SW0-T7-3				
	M-SW3-T7	M-SW3-T7-1		3	9	291
		M-SW3-T7-2				
		M-SW3-T7-3				
	M-SW6-T7	M-SW6-T7-1		6	18	282
		M-SW6-T7-2				
		M-SW6-T7-3				
	M-SW9-T7	M-SW9-T7-1		9	27	273
		M-SW9-T7-2				
		M-SW9-T7-3				
	M-SW12-T7	M-SW12-T7-1		12	36	264
		M-SW12-T7-2				
		M-SW12-T7-3				
T14	M-SW0-T14	M-SW0-T14-1	14	0	0	300
		M-SW0-T14-2				
		M-SW0-T14-3				
	M-SW3-T14	M-SW3-T14-1		3	9	291
		M-SW3-T14-2				
		M-SW3-T14-3				
	M-SW6-T14	M-SW6-T14-1		6	18	282
		M-SW6-T14-2				
		M-SW6-T14-3				
	M-SW9-T14	M-SW9-T14-1		9	27	273
		M-SW9-T14-2				
		M-SW9-T14-3				
	M-SW12-T14	M-SW12-T14-1		12	36	264
		M-SW12-T14-2				
		M-SW12-T14-3				
T28	M-SW0-T28	M-SW0-T28-1	28	0	0	300
		M-SW0-T28-2				
		M-SW0-T28-3				
	M-SW3-T28	M-SW3-T28-1		3	9	291
		M-SW3-T28-2				
		M-SW3-T28-3				
	M-SW6-T28	M-SW6-T28-1		6	18	282
		M-SW6-T28-2				
		M-SW6-T28-3				
	M-SW9-T28	M-SW9-T28-1		9	27	273
		M-SW9-T28-2				
		M-SW9-T28-3				
	M-SW12-T28	M-SW12-T28-1		12	36	264
		M-SW12-T28-2				
		M-SW12-T28-3				

^1^ D means the dosage of OSP.

**Table 2 materials-14-05433-t002:** Mix proportion of magnesium phosphate cement (MPC) (g/cm^3^).

MO ^1^	KH_2_PO_4_	Borax	FA	Water
1.17	0.80	0.12	0.30	0.34

^1^ MO is the blend of MgO and OSP.

**Table 3 materials-14-05433-t003:** Chemical components of MgO and fly ash (FA).

Mass Content (%)	MgO	TiO_2_	SO_3_	Al₂O₃	Fe₂O₃	CaO	SiO₂	LOI
MgO	96.25	/	/	0.29	1.09	1.18	1.16	0.03
FA	3.3	1.86	0.8	24.58	6.55	4.87	56.47	1.3

**Table 4 materials-14-05433-t004:** Chemical components of oyster shell powder (OSP).

Components	CaCO_3_	H_2_O	Ba	As	Cd	Pb
Mass Content (%)	97.79	0.71	0.03	0.0001	0.00002	0.001

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
