# Peer review of "A New Magnesium Phosphate Cement Based on Renewable Oyster Shell Powder: Flexural Properties at Different Curing Times"

_materials, 2021, doi:10.3390/ma14185433_

Round 1

Reviewer 1 Report

This paper focuses on the influence of different oyster shell powder mass contents on a new magnesium phosphate cement flexural property. The purpose of this paper is to discuss an important issue for expanding use of magnesium phosphate cement and promoting the recycling of resources. However, I think that the results are not well discussed and conclusions are not meaningful in paper.

There are some insufficient explanations in this paper. For example, it is difficult to discuss the existence of pore detects and microcracks in the matrix using only SEM image (Fig.10). I think it is necessary to include the results of pore size distribution of the matrix. And, the authors need to describe the detailed explanations of XRD measurement conditions and sample preparation methods. Furthermore, it is difficult to evaluate the degree of brittleness by using only failure mode (Fig.5). In order to evaluate the brittleness of specimen, the authors should consider using the tension softening curve and fracture energy of L-D curve.

Author Response

Thank you for your valuable comments and suggestions. We have made the following explanations and answers to your questions, hoping to answer your questions.

The purpose of this paper is to discuss an important issue for expanding oyster shell powder’s use in magnesium phosphate cement and promoting the recycling of resources.And oyster shell powder is renewable resources,adding OSP to MPC to partially replace magnesium oxide,promoting the recycling of resources.

In Section 5,cross-sections analysis are added to discuss the existence of pore detects and microcracks.

The detailed explanations of XRD measurement conditions and sample preparation methods are added in Section 2.1.

In Section 4.1,the peak displacement and secant stiffness analysis of the specimen is to evaluate the brittleness of specimen,and the tension softening curve and fracture energy of L-D curve need further study.

Reviewer 2 Report

Manuscript ID: materials-1348568
Type of manuscript: Article
Title: A new magnesium phosphate cement based on renewable oyster shell
powder: flexural properties at different curing ages
Authors: Hui Wu, Zhujian Xie, Liwen Zhang *, Zhiwei Lin, Shimin Wang, Wenle
Tang

The paper deals with a new binder consisting of potassium dihydrogen phosphate (KH2PO4), borax, fly ash (FA), MgO and a certain amount of substituting OSP for MgO. All ingredients are fine-grained powders. The authors should present a particle size distribution for all the powders.

The reviewer’s general comment refers to the sample preparation procedure of samples without coarse-grained filler. What the standards were used for sample preparation? Why the Authors did not prepare the mortars, i.e. filler (e.g. sand) and the binder proposed in this work according to the standard procedure. This approach make possible to compare the results obtained with the presented by other researchers.

Table 4 presents both chemical and phase composition. How the CaCO3 content was determined?

There is no results supporting the proposed mechanism in the Figs. 12-14. The way in presenting the XRD results make it very difficult to track the reaction degree.

The maximum intensity in Fig. 11 – the unit presented is probably counts ?

The results presented should be supported with other, e.g. thermal analysis DTA/DSC-TG-EGA or FT-IR. Thus, it will be possible to determine the possible amorphous phase.

Taking into account the scientific aspect of this work, I cannot recommend this work in the presented form.

Author Response

Thank you for your valuable comments and suggestions. We have made the following explanations and answers to your questions, hoping to answer your questions.

A particle size distribution for all the powders has presented in Section 2.2.

MPC is generally by magnesium oxide (MgO), potassium dihydrogen phosphate (KH2PO4) and retarder and other additives mixed in a certain proportion, acid and base neutralization reaction formed an environmentally friendly cementite material[1-4].It’s not traditional cement mortar and its preparation procedure of samples is the same as preliminary study[5-7].The raw materials used and the manufacturing process of specimens in this paper are in Section 2.1.

The CaCO3 content presented in Table 4 was provided by Merchant detection, Lingshou County Ruixing mineral powder factory,ShiJiaZhuang City, HeBeiProvince,China.

During the preparation of the specime,after the exciting of water, MgO undergoes a hydration together with phosphate, combining to produce a new compound named potassium phosphate magnesium (K-struvite) around unreacted MgO [8], according to the process shown in Fig. 12,with the incorporation of OSP,the unit matrix substrate has changed ,the process shown in Fig. 14.And by XRD analysis of OSP-MPC, new hydration products were generated, and the elements in hydration products were traced to obtain the chemical reaction equation,as shown in the fig.13

Hydration reaction in MPC is a continuous and lasting process, and hydration reaction will continue to occur in the MPC after the 28-day curing period. Therefore, in this paper, The hydration product MKP is analyzed by XRD to determine the process of hydration reaction.

The unit of maximum intensity in Fig. 11 means arbitrary unit.Since absolute intensity is meaningless, it can be expressed in terms of relative relative intensity.

Our team is working on the MPC thermal analysis TG-DSC,in this paper, the microstructure and chemical composition of MPC were analyzed by SEM and XRD.

References:

  • Zhu Ding,Biqin Dong,Feng Xing,Ningxu Han,Zongjin Li. Cementing mechanism ofpotassium phosphate based magnesium phosphate cement[J]. Ceramics International,2012,38(8):
  • Yu Jincheng,Lin Ludan,Qian Jueshi,Jia Xingwen,Wang Fan. Preparation and properties of a low-cost magnesium phosphate cement with the industrial by-products boron muds[J]. Construction and Building Materials,2021,302:
  • Huan Zhou,Anand K. Agarwal,Vijay K. Goel,Sarit B. Bhaduri. Microwave assisted preparation of magnesium phosphate cement (MPC) for orthopedic applications: A novel solution to the exothermicity problem[J]. Materials Science & Engineering C,2013,33(7):
  • Yue Li,Tongfei Shi,Bing Chen,Yaqiang Li. Performance of magnesium phosphate cement at elevated temperatures[J]. Construction and Building Materials,2015,91:Figure 11 - a.U. is a common unit for XRD detection
  • Zhang Li-wen, Jiang Zu-qian, Wu Hui, et al. Flexural Properties of Renewable Coir Fiber Reinforced Magnesium Phosphate Cement, Considering Fiber Length[J]. Materials, 2020, 13(17).
  • Zhang, Li-wen, Jiang, Zu-qian, Zhang Wen-hua, et al. Flexural Properties and Microstructure Mechanisms of Renewable Coir-Fiber-Reinforced Magnesium Phosphate Cement-Based Composite Considering Curing Ages[J]. Polymers, 12(11).
  • Jiang Zu qian, Zhang Li-wen, Geng Tao, et al. Study on the Compressive Properties of Magnesium Phosphate Cement Mixing with Eco-Friendly Coir Fiber Considering Fiber Length[J]. Materials, 2020, 13(14) 
  • Xu, B.; Lothenbach, B.; Leemann, A.; Winnefeld, F. Reaction mechanism of magnesium potassium phosphate cement with high magnesium-to-phosphate ratio. Cement Concrete Res. 2018, 108, 140-151.

Reviewer 3 Report

This work presents a study on the use of ground oyster shell powder to replace cement in the magnesium phosphate cementitious system. The idea is interesting, but still some improvement to be done as follows:

  1. You should make a cost analysis when using OSP, should this involved in an intensive grinding process to obtain OSP.
  2. Better to describe the physical, chemical properties of all raw materials in the section 1.
  3. Lines 85-88: what do you mean by failed specimens? If that are the samples from mechanical tests, then it is not the right way to prepare samples for SEM/EDX measurements. There will be microcracks formed which are not representative for the original materials.
  4. I don’t see much added values of Figs. 5 and 6.
  5. 7: are these figures plotted from measurement data?
  6. SEM analysis should be extended, not just showing the SEM images. I also see a lot of microcracks, which are the indications of improper preparation of SEM samples (check here, for example [1]).
  7. Lines 236-238: you need to describe sample preparation for XRD.
  8. Please explain clearly the proposed reaction model shown in Fig. 14.
  9. Basically, OSP has similar compositions and properties like limestone filler. Should you compare your work with other published works using limestone filler to replace cement.
  10. Language should be improved.

Reference.

  1. Phung, Q.T., et al., Investigation of the changes in microstructure and transport properties of leached cement pastes accounting for mix composition. Cement and Concrete Research, 2015. 79: p. 217-234.

Author Response

Thank you for your valuable comments and suggestions. We have made the following explanations and answers to your questions, hoping to answer your questions.

  • Preparation process of OSPwas shown in fig.4. Firstly, the sludge and water grass attached to the surface of oyster shell are washed and dried naturally for preliminary crushing. The preliminary crushing process is carried out through laboratory jaw crusher.Then, the discarded shell fragments after preliminary crushing were followed by a light drum ball mill and an experimental ball mill for primary and secondary grinding. Finally, the OSP was screened to a finness of about 325 mesh with a residual of less than 6% through a vibrating screen.
  • The physical, chemical properties of all raw materials had added in the section 2.2.
  • Under the action of the three-point bending test, the mid-span fracture of the specimen was observed, while the samples prepared for SEM/EDX measurement were selected in the non-mid-span failure region, so there would be no additional micro-cracks on the sample due to mid-span fracture.
  • It can be intuitively observed in Figure 5 that all the failure modes of specimens are brittle failure, and the impact of OSP production on the crack opening width and brittleness of MPC specimens under different curing ages is shown from the side of crack opening width.In Figure 6, the improvement effect of OSP dosage on anti-folding performance and the ability to resist crack expansion are reflected by the degree of smoothness and concave-convex degree of fracture section of the specimen and the distribution of the number of pores in the interface. For detailsin Section 3.
  • Yes,these figuresare plotted from measurement data.If you need to check the experimental data, I can send it to you.
  • The SEM analysis in this paper mainly focuses on the changes of the microstructure of MPC under different curing ages and different dosages of OSP, mainly observing the crystal morphology changes of internal components and the rough distribution of internal cracks. The changes of MPC performance are analyzed by combining SEM images with the pore distribution on the fracture surface of specimens,as shown in Section 5.
  • The detailed explanations of XRD measurement conditions and sample preparation methodsare added in Section 2.1.
  • During the preparation of the specime,after the exciting of water,MgO undergoes a hydration together with phosphate, combining to produce a new compound named potassium phosphate magnesium (K-struvite) around unreacted MgO[1], according to the process shown in Fig.12,with the incorporation of OSP,the unit matrix substrate has changed ,the process shown in Fig. 
  • Limestone filler and OSP have similar main components. It is a good idea to consider the influence of limestone filler and OSP on their performance by taking magnesium phosphate cement as the matrix.But because of other MPC raw material mix ratio, production method, admixture and other factors,it’s difficult to compare our work with other published works using limestone filler to replace cement.And we will take this suggestion into account in the future research direction to replace OSP with limestone for demonstration tests.
  • Thank you for your reminding. We have improved our language under the guidance of professional teachers

References:

[1] Xu, B.; Lothenbach, B.; Leemann, A.; Winnefeld, F. Reaction mechanism of magnesium potassium phosphate cement with high magnesium-to-phosphate ratio. Cement Concrete Res. 2018, 108, 140-151.

Round 2

Reviewer 1 Report

1. p.8 Line 166, p.11 Line 236

It is necessary to discuss the brittleness of the specimen based on the parameters such as fracture energy and tension softening curve in order to evaluate the brittleness. The expression “brittleness” should be deleted in this paper.

2. p.10, Figure.9

The two figures ((c)and (d)) in the bottom row of Figure.9 are duplicates. Please delete them.

3. p.11, Line 242-245

Please explain the method of cross sections analysis in detail. And the authors should show and discuss the data from the analysis.

Author Response

Thank you very much for your comments and suggestions. We have further explored the previous questions and made the following replies, and made appropriate adjustments in the article.

1. The expression “brittleness” has been deleted in reference to your comments,And the meaning of brittleness of concrete is deeply understood.

2. Duplicate figures have been deleted.thankyoufor the reminder.

3. Image analysis technology was used to study the holearea of fracture section of specimens with different DOSages of OSP in different curing periods. The image analysis method and analysis of the obtained data are detailed in Section 3.

Reviewer 2 Report

I accept the corrected version of this paper.

Author Response

Thank you for your recognition and affirmation of our work

Reviewer 3 Report

Not much improved compared to the previous version. I don't this manuscript is good enough to be published at this state. 

Author Response

Thank you very much for your comments and suggestions. We have further explored the previous questions and made the following replies, and made appropriate adjustments in the article.

  • Thank you for your suggestion. We add waste oyster shells to MPC through processing and grinding processes in order to recycle the excess oyster shell resources and reduce the cost of MPC by partially replacing magnesium oxide。Preparation process of OSPwas shown in fig.4. Firstly, the sludge and water grass attached to the surface of oyster shell are washed and dried naturally for preliminary crushing. The preliminary crushing process is carried out through laboratory jaw crusher.Then, the discarded shell fragments after preliminary crushing were followed by a light drum ball mill and an experimental ball mill for primary and secondary grinding. Finally, the OSP was screened to a finness of about 325 mesh with a residual of less than 6% through a vibrating screen.
  • In section 1(Introduction), the kinds of raw materials needed for the preparation of MPC and the main components and properties of OSP are briefly introduced. And the physical, chemical properties of all raw materials had been detialedin the section 2.2.
  • Under the action of the three-point bending test, the mid-span fracture of the specimen was observed, while the samples prepared for SEM/EDX measurement were selected in the non-mid-span failure region, so there would be no additional micro-cracks on the sample due to mid-span fracture.However, the cracks caused by mistakes in the preparation of XRD samples will be screened out after the image is obtained
  • 5 (a), (b) and (c) respectively manifest specimens’ failure modes under three curing ages, which displays an identical failure mode for specimens in each group when the load was added to the peak load of flexural strength.The Fig. 6 (a), (b) and (c) respectively show the cross-interruption surface of the specimen after its failure was intercepted,And provide the basis for the following image analysis work,see Section 3for details.
  • Yes,these figuresare plotted from measurement data.
  • The SEM analysis in this paper mainly focuses on the changes of the microstructure of MPC under different curing ages and different dosages of OSP, mainly observing the crystal morphology changes of internal components and the rough distribution of internal cracks. The changes of MPC performance are analyzed by combining SEM images with the pore distribution on the fracture surface of specimens,while the sample size of SEM is about 2mm*2mm, it is difficult to avoid the occurrence of sample edge in the process of sample preparation as shown in Section 5.
  • The detailed explanations of XRD measurement conditions and sample preparation methodsare added in Section 2.1
  • During the preparation of the specime,after the exciting of water,MgO undergoes a hydration together with phosphate, combining to produce a new compound named potassium phosphate magnesium (K-struvite) around unreacted MgO[1], according to the process shown in Fig.12,with the incorporation of OSP,OSP partially replaces magnesium oxide,the unit matrix substrate has changed ,the process shown in Fig. 
  • According to your suggestion, we have added limestone to replace magnesium oxide in the article and made a simple comparison with OSP to partially replace magnesium oxide in this article. Limestone and OSP have similar composition and performance, resulting in similar effects on MPC performance. Please refer to Section 6 for details.
  • Thank you for your reminding. We have polished up the language of the article.

References:

[1] Xu, B.; Lothenbach, B.; Leemann, A.; Winnefeld, F. Reaction mechanism of magnesium potassium phosphate cement with high magnesium-to-phosphate ratio. Cement Concrete Res. 2018, 108, 140-151.

Round 3

Reviewer 1 Report

p.7 Line 153-158

The details of the image analysis techniques are unknown. You need to explain the number of pixels, resolution, measurement area size, and so on. And please indicate the measurement range (void diameter) that can be identified in this image analysis technique.

p.8 Fig.7

“mm2”on the vertical axis should be replaced by a %, such as the ratio of the hole area to the measurement area.

Author Response

Thank you for your comments and suggestions. We have improved and supplemented the article based on your comments.

1、We supplement the details on image analysis techniques, as described in Section 3.

2、Thank you for your suggestion.We have replaced the hole area (mm2)with the area ratio(%). Details shown in Section 3.

Reviewer 3 Report

English should be improved before publication.

Author Response

Thanks for your suggestion, we have improved some shortcomings of the article language